# Chemo-Enzymatic Production of 4-Nitrophenyl-2-acetamido-2-deoxy-α-D-galactopyranoside Using Immobilized β-*N*-Acetylhexosaminidase

Helena Hronská [1,*], Vladimír Štefuca [1], Ema Ondrejková [1], Mária Bláhová [1], Jozef Višňovský [2] and Michal Rosenberg [1]

[1] Institute of Biotechnology, Faculty of Chemical and Food Technology, Slovak University of Technology, Radlinského 9, 812 37 Bratislava, Slovakia; vladimir.stefuca@stuba.sk (V.Š.); ema.ondrejkova@stuba.sk (E.O.); maria.blahova@stuba.sk (M.B.); michal.rosenberg@stuba.sk (M.R.)

[2] SynthCluster, s.r.o., Moyzesova 15, 900 01 Modra, Slovakia; jozef.visnovsky@gmail.com

* Correspondence: helena.hronska@stuba.sk; Tel.: +421-259-325-713

**Abstract:** α-Nitrophenyl derivatives of glycosides are convenient substrates used to detect and characterize α-*N*-acetylgalactosaminidase. A new procedure combining chemical and biocatalytic steps was developed to prepare 4-nitrophenyl-2-acetamido-2-deoxy-α-D-galactopyranoside (4NP-α-GalNAc). The α-anomer was prepared through chemical synthesis of an anomeric mixture followed by selective removal of the β-anomer using specific enzymatic hydrolysis. Fungal β-*N*-acetylhexosaminidase (Hex) from *Penicillium oxalicum* CCF 1959 served this purpose owing to its high chemo-and regioselectivity towards the β-anomeric *N*-acetylgalactosamine (GalNAc) derivative. The kinetic measurements of the hydrolytic reaction showed that the enzyme was not inhibited by the substrate or reaction products. The immobilization of Hex in lens-shaped polyvinyl alcohol hydrogel capsules provided a biocatalyst with very good storage and operational stability. The immobilized Hex retained 97% of the initial activity after ten repeated uses and 90% of the initial activity after 18 months of storage at 4 °C. Immobilization inactivated 65% of the enzyme activity. However, the effectiveness factor and kinetic and mass transfer phenomena approached unity indicating negligible mass transfer limitations.

**Keywords:** anomer separation; *N*-acetylhexosaminidase; glycoside; chromogenic enzyme substrates; enzyme immobilization

## 1. Introduction

Glycosides are important bioactive molecules with wide distribution in nature. They play critical roles in many biological processes and strongly influence human health [1,2]. One of the many practical applications is their use as effective diagnostic tools in clinical microbiology [3]. Synthetic glycosides with chromogenic or fluorogenic groups are often used as substrates for testing hydrolytic enzymes [4]. Initially, nitrophenylglycosides were frequently used as substrates of carbohydrases [5]. The main advantages of chromogenic and fluorogenic substrates are their high reactivity and sensitivity, which enable the rapid spectrophotometric detection of target enzymes and enzyme-based assays [6]. However, the traditional chemical synthesis of glycosides based on the Koenigs-Knorr method and its modifications is complicated, inefficient, financially demanding, and not environmentally friendly. Moreover, to obtain anomerically pure compounds, various protecting groups, harsh conditions, and often toxic reagents are needed [7]. These conditions prove to be problematic when the prepared compounds are used in the food or pharmaceutical industries [7,8]. Therefore, the current trend is to replace chemical approaches with more efficient, cheaper, and ecological enzymatic methods.

One of the possible methods is a chemo-enzymatic process starting with anomerically nonselective acid-catalyzed glycosylation. The resulting mixture of anomeric products

can be used for preparing the desired glycosidic anomer by enzymatic hydrolysis of the undesired anomer. This way, many simple glycosides have been prepared, although the anomeric purity depends on the type of glycoside and enzyme used [8–10]. In this sense, fungal β-*N*-acetylhexosaminidases (EC 3.2.1.52; CAZy GH20; http://www.cazy.org, accessed 1 January 2020; Hex) have the hydrolytic potential in preparing pure α-nitrophenyl derivatives of glycosides. Hex play an essential role in chitin degradation but also in other biological processes [11]. They belong to a broad group of glycoside hydrolases. Besides the hydrolytic activity, under certain reaction conditions, Hex can catalyze transglycosylation reactions, even for structurally modified substrates [12,13]. The synthetic potential of Hex from prokaryotes and eukaryotes was intensively studied resulting in the preparation of the glycosidic products such as immunoactive oligosaccharides [14,15] or more complex glycoconjugates [16]. Various fungal producers of β-*N*-acetylhexosaminidases have been studied in recent years. The best and currently used strains are *Penicillium oxalicum* [17], *Aspergillus oryzae* [18], and *Talaromyces flavus* [19].

Among the various enzymatic substrates in form of α-nitrophenyl derivatives of glycosides, 4-Nitrophenyl-2-acetamido-2-deoxy-α-D-galactopyranoside (4NP-α-GalNAc) is used to determine the activity of α-*N*-acetylgalactosaminidase (EC.3.2.1.49, α-NAGA), an exo-glycosidase, which hydrolyzes the α-*N*-acetyl-D-galactosamine bond of the glycoconjugates [20]. The α-NAGA enzyme has been isolated from different eukaryotic and prokaryotic sources and its potential for application in the medical field has increased during the past decade. α-NAGA is recommended for the enzymatic transformation of red blood cell group A to 0 cells. Completely removing the A blood group antigen from red blood cells would alleviate the blood group 0 reservoir shortage in blood banks [21].

Moreover, α-NAGA is a crucial enzyme involved in a rare inherited metabolic disorder (Schindler disease) that primarily causes neurological problems. The three existing forms of Schindler disease are characterized by low levels or deficient activity of the lysosomal enzyme α-NAGA. The enzymatic defect leads to the abnormal accumulation of glycopeptides and oligosaccharides with α-*N*-acetylgalactosaminyl residues in specific tissues in the human body and urine. Schindler disease type I is the most severe form; people usually do not survive early childhood. There is no specific therapy for patients with this disorder. Early diagnosis based on the determination of α-NAGA level may benefit the affected individuals and their families. The enzyme replacement therapy or gene therapy is considered a possible treatment approach for lysosomal disorders [22,23].

The interest in practical applications of hexosaminidases has resulted in efforts to immobilize them using different supports and methods. For example, Hex from *Turbo cornutus* was immobilized by crosslinking with BSA and glutaraldehyde [24], plant Hex was covalently coupled to paramagnetic beads [25], bacterial His-tagged Hex was specifically linked to agarose beads [15], and human Hex was immobilized via glutaraldehyde on polylactic acid films [26]. However, according to our literature search, the immobilization of fungal Hex has not been mentioned so far.

This paper describes the preparation of 4-nitrophenyl-2-acetamido-2-deoxy-α-D-galactopyranoside, the substrate of α-NAGA. The process begins with the chemical synthesis of an anomeric mixture of α/β-substituted *N*-acetylgalactosamine (GalNAc) derivatives. Subsequently, the α-anomer is purified by the selective hydrolysis of the β-anomer to nitrophenolic aglycon and glycosidic residue GalNAc using Hex. The physicochemical properties of the released β-GalNAc and nitrophenol are different from the remaining α-anomer of the glycoside, and they are, therefore, easily separated by the proposed two-step extraction.

Fungal Hex was used in both the free and immobilized forms. The application of immobilized Hex for the hydrolysis of anomeric mixtures of nitrophenyl glycosides has not been reported. In this work, the enzyme was immobilized in lens-shaped polyvinyl alcohol hydrogel particles for repeated hydrolysis runs of β-nitrophenyl derivatives of glycosides.

## 2. Results and Discussion

A simple and convenient procedure was developed to synthesize α-anomer of nitrophenyl galactoside derivative using free or immobilized β-*N*-acetylhexosaminidase. In this reaction system, α/β-substituted GalNAc derivatives were chemically synthesized and the β-substituted GalNAc derivatives were hydrolyzed, thus allowing for the easy isolation of the α-substituted GalNAc derivatives (Figure 1).

**Figure 1.** Hydrolysis of mixtures of α/β-anomeric substituted galactopyranosides using β-*N*-acetylhexosaminidases. (**1**) 4NP-β-GalNAc, (**2**) 4NP-α-GalNAc, (**3**) 4-nitrophenol, (**4**) GalNAc.

A fungal β-*N*-acetylhexosaminidase (hexosaminidase, E.C. 3.2.1.52) from *Penicillium oxalicum* was used for the selective hydrolysis of the β-anomeric GalNAc derivatives. The strains of *Penicillium oxalicum* exhibited 2.5-fold higher GalNAcase activity compared with other fungal producers, e.g., *Aspergillus oryzae* or *Fusarium oxysporum* [27].

### 2.1. β-N-Acetylhexosaminidase Production

The production and purification of extracellular β-*N*-acetylhexosaminidases from *P. oxalicum* CCF 1959 have been previously described in detail [17,28]. The maximum enzyme activity was achieved after 12–13 days of cultivation at 28 °C (typically 0.3–0.4 U/mL of culture medium). The enzyme was purified to apparent SDS-PAGE homogeneity by cation exchange chromatography at pH 3.5 before being concentrated and stored in 1 M ammonium sulfate at 4 °C. The specific activity of the purified enzyme towards 4NP-β-GalNAc was 300–400 U/mg protein. The decrease in the activity of the enzyme preparations stored at 4 °C after six months was not more than 5% of the original activity.

### 2.2. Enzyme Hydrolysis of Nitrophenyl Derivatives of β-GalNAc

Chromogenic substrates are generally used in rapid diagnostics or for the determination of kinetic parameters of specific enzymatic reactions [29]. Monitoring the initial reaction rate is sufficient for such purposes. However, the main goal of this work was to achieve the highest possible purity of the α-anomer by complete hydrolysis of the β-anomer of nitrophenyl glycoside. Therefore, the hydrolysis of the β-anomer of 4-NPGalNAc was investigated at various initial concentrations. The hydrolysis proceeded until complete conversion and was independent of the initial substrate concentration (Figure 2). Simple Michaelis–Menten kinetics (Equation (1)) was applied to the data, and the kinetic parameters ($V_{max}$ = 0.4560 ± 0.0075 mM/min and $K_M$ = 6.50 ± 0.16 mM) were determined. The value of $k_2$ (Equation (2)) corresponding to an enzyme concentration of 30.7 mg/L was 0.0149 mmol/min/mg. The kinetic data demonstrated that the substrate or the reaction product did not inhibit the enzyme. The absence of product inhibition effect that is always enhanced by mass transfer limitations increasing the particle product concentration was a good prerequisite for the enzyme immobilization by entrapment.

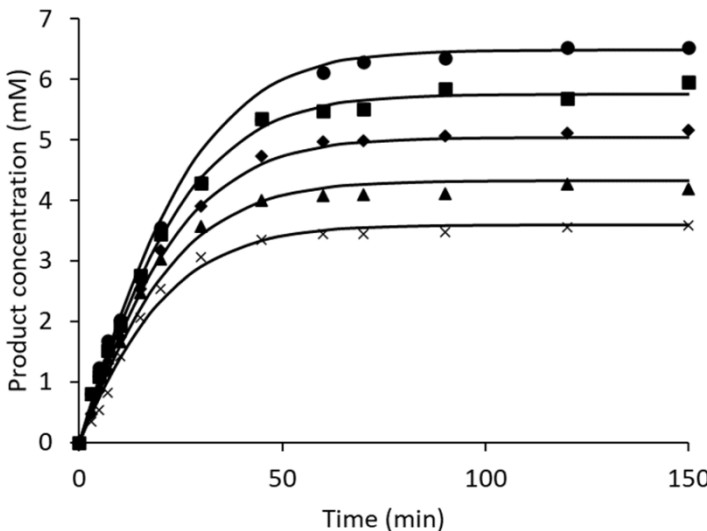

**Figure 2.** Kinetics of the enzymatic hydrolysis of 4-NP-β-GalNAc. Initial substrate concentrations in mM: × 3.6; ▲ 4.32; ◆ 5.04; ■ 5.76; ● 6.48. Solid lines represent data calculated from the Michaelis–Menten kinetic model.

### 2.3. Chemo-Enzymatic Production of the α-Anomer from the Anomeric Mixture of 4-NP-α/β-GalNAc Derivatives

Nonselective glycosylation produces anomeric mixtures of glycosides while the equilibrium ratio of the α-and β-anomers depends on the thermodynamic stability of the individual isomers [30]. Their separation by traditional methods such as recrystallization or chromatography from the reaction mixture is relatively tricky from a technical and economic point of view [8]. The enzyme reaction-based approaches are promising methods for producing and separating desired anomers from anomeric mixtures of glycosides. In the present study, the β-anomer of nitrophenyl glycosides was selectively hydrolyzed using fungal hexosaminidases; this allowed the subsequent purification of the α-anomer by extraction techniques. Substrates were aqueous (4 mM) solutions of anomeric mixtures prepared from standard chemicals (anomeric ratio of α:β = 1:1). Subsequently, the activity of Hex was easily determined by measuring the time dependence of 4-nitrophenol concentration in the reaction mixture. Hydrolytic reactions were performed under the same conditions as in previous experiments. The Hex showed high specificity and activity towards the β-anomer of 4-NPGalNAc. Under the given conditions, the degree of decomposition of the β-anomer was greater than 96% after 60 min of reaction. Finally, all the α-anomer of 4-NPGalNAc and the hydrolysis products of the β-anomer (4-nitrophenol and 2-acetamido-2-deoxy-D-galactose) remained in the reaction mixture.

### 2.4. Separation of Products

The α-anomer can usually be isolated from the resulting mixture by tedious chromatographic procedures reported in the literature [31]. Here, a simple method is presented to separate the individual components from the reaction mixture. The compounds resulting from the hydrolysis of the β-anomer (4-nitrophenol and *N*-acetylgalactosamine) have different physical properties compared to the remaining α-anomer of the glycoside; thus, they are easily separated from the resulting mixture and reused for a glycoside synthesis. This approach is similar to a typical example of cyclic cascade designs where one enantiomer of a racemic mixture is selectively removed by hydrolysis to the stable products; in turn, the products are returned to the start of the chemical synthesis of the racemic mixture [32]. According to this, after hydrolysis, the reaction mixture was treated by extraction with two organic solvents. In the first step, 4-nitrophenol was extracted using ethyl acetate. Then, 4-NP-α-GalNAc was extracted from the aqueous phase by n-butanol resulting in 98.5% of the total yield. The butanol extract was concentrated under reduced pressure and then

crystallized with the addition of methanol. The yield of 4-NP-α-GalNAc in crystals was 90.5%. *N*-acetyl-D-galactosamine remained in the aqueous phase. In the case of the expensive aglycones or β-glycones, this extraction technique allows their efficient recycling. The optical purity of the crystals of 4-nitrophenyl-2-acetamido-2-deoxy-α-D-galactopyranoside was higher than 99.6% and the identity of the substance was also confirmed by NMR analysis (1H NMR (CDCl3) α-anomer 5.60 ppm (d, J = 3.35 Hz); β-anomer 5.14 ppm (d, J = 8.41 Hz)).

### 2.5. Enzyme Hydrolysis in Crystal Suspensions of 4-NP-α/β-GalNAc

The low initial concentration of the anomeric mixture due to the very low solubility of the aromatic glycosides in an aqueous solution limits the final product titer and volumetric productivity. To overcome this problem, enzyme hydrolysis was applied directly on crystal suspensions of α/β-mixtures of aromatic glycosides. Here, the corresponding α-anomer remained in the reaction mixture mostly in a crystal form while the β-anomer was completely hydrolyzed. The reaction mixture with a higher 4-NP-α/β-GalNAc concentration (100 mM, anomeric ratio of α:β = 2:3) was prepared by the chemical synthesis explained in the Materials and Methods section. All samples were analyzed using HPLC, whereby the concentration of dissolved substances in the reaction mixture was monitored (Figure 3). The β-anomeric crystals were gradually dissolved in the reaction mixture during the hydrolysis by Hex. The concentration of the β-anomer in the reaction mixture decreased, while the concentration of the dissolved α-anomer stabilized after a short time (the rest remained in the form of crystals). Simultaneously, the concentration of 4-nitrophenol increased, corresponding to the total amount of the hydrolyzed β-anomer.

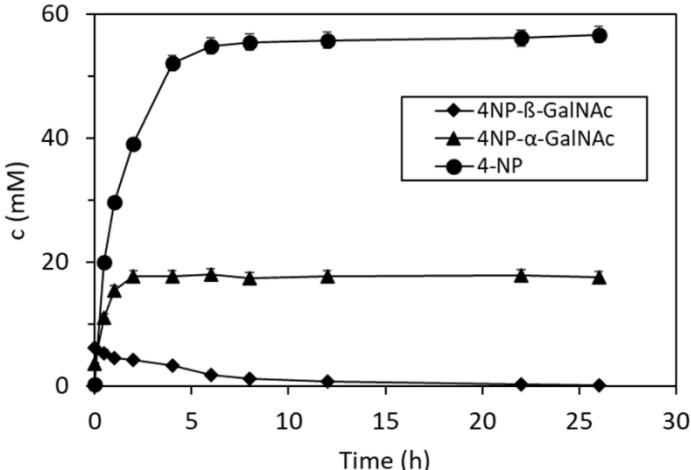

**Figure 3.** Hydrolysis of the β-anomer in a chemically prepared anomeric mixture of 4-NP-α/β-GalNAc. Reaction conditions: 30 mL 100 mM 4-NP-α/β-GalNAc, protein concentration 3 μg/mL, pH 4.5; temperature 35 °C, 300 rpm.

After 6 h of enzyme hydrolysis of 100 mM 4-NP-α/β-GalNAc, the α-anomer of 4-NPGalNAc was separated from the reaction mixture. Initially, the crystalline α-anomer was filtered off from the remaining mixture and dried in the oven for 30 min at 50 °C. The purity determined by HPLC was 96%. After re-solubilization of the α-anomeric crystals, a small amount of β-anomer was also present (for more details, see the HPLC record Figure S1). It was assumed that some part of β-anomer remained "enclosed" in substrate crystals due to their insufficient crushing in the grinding mortar before the reaction, preventing its complete hydrolysis.

### 2.6. Substrate Specificity of β-N-Acetylhexosaminidase

The ability of the currently studied Hex to act on different β-anomers of glycoside derivatives was investigated by hydrolysis of five selected β-substrates under standard

assay conditions (Table 1). The preference of Hex for an axial (GalNAc) or equatorial (GlcNAc) 4-OH group on the substrate pyranozyl ring depends on the source of the enzyme [27]. The enzyme most effectively hydrolyzed 4-NP-β-GalNAc and 2-nitrophenyl-2-acetamido-2-deoxy-β-D-galactopyranoside (2-NP-β-GalNAc); these results were similar to those of Weignerová et al. who studied the substrate specificity of Hex from various fungal sources [27]. The enzyme activity of Hex is approximately similar for nitrophenyl derivatives of GalNAc (422.2 U/mg for 4-NP-β-GalNAc and 416.7 U/mg for 2-NP-β-GalNAc). The substrates with an equatorial 4-OH group (4-nitrophenyl-2-acetamido-2-deoxy-β-D-glucopyranoside (4-NP-β-GlcNAc) and 2-nitrophenyl-2-acetamido-2-deoxy-β-D-glucopyranoside (2-NP-β-GlcNAc)) and the fluorogenic substrate (4-methylumbelliferyl-2-acetamido-2-deoxy-β-D-galactopyranoside(4-MU-β-GalNAc)) were hydrolyzed to a lesser extent (Table 1). The enzymatic hydrolysis of more than 97% of all substrates was completed in a maximum of 60 min under the given experimental conditions.

**Table 1.** Substrate specificity of Hex from *Penicillium oxalicum* CCF 1959.

| Substrate | Specific Enzyme Activity (U/mg) $\pm$ SD |
|---|---|
| 4NP-β-GalNAc | 422 $\pm$ 17 |
| 4NP-β-GlcNAc | 135.3 $\pm$ 5.5 |
| 2NP-β-GalNAc | 417 $\pm$ 17 |
| 2NP-β-GlcNAc | 156.6 $\pm$ 6.7 |
| 4MU-β-GalNAc | 97.1 $\pm$ 4.6 |

*2.7. Immobilization of β-N-Acetylhexosaminidase*

Immobilization of the enzyme has some undeniable advantages compared to using a free enzyme. These include reusability, enhanced stability, easy handling, convenient separation by filtration from the reaction mixture, and long-term storage. However, immobilization can cause changes in enzyme activity. It is, therefore, necessary to study the efficiency of immobilization, specific activity, and long-term stability of the immobilized enzymes.

β-*N*-acetylhexosaminidase isolated from the fungal strain *P. oxalicum* CCF 1959 was immobilized into lens-shaped PVA hydrogel capsules. The hydrolysis of 4-NP-β-GalNAc catalyzed by the immobilized enzyme is depicted in Figure 4; also included are the data resulting from mathematical modeling after considering limitations of particle mass transfer. Here, the $K_M$ value of the soluble enzyme (6.5 mM) was applied while the value of the catalytic constant $k_2$ was determined using the experimental data by the least-squares method. The obtained value of $k_2$ = 0.00513 mmol/min/mg was 34.5% of that for the enzyme before immobilization. Thus, 65.5% of the enzyme activity was lost by enzyme inactivation and/or by rendering the enzyme inaccessible for the substrate by gel entrapment. A similar effect was observed previously wherein glycosidases were used as the biocatalyst [33,34]. The effectiveness factor values for mass transfer were above 0.99 during the entire reaction, indicating nearly negligible mass transfer limitations.

*2.8. Reusability and Storage Stability of the Immobilized Enzyme*

The repeated usability of immobilized biocatalysts is a crucial parameter in determining their potential industrial application. The reusability and long-term stability of the immobilized Hex were determined in the repeated batch mode of operation. Two types of immobilizates with different protein concentrations were prepared. Type A: 10.0 μg prot./g PVA gel and type B: 15.9 μg prot./g PVA gel were tested in repeated hydrolysis of 4-NP-β-GalNAc (4 mM). Preparation B (higher immobilized amount) demonstrated a lower specific activity (40 U/mg prot.) than that observed for preparation A (45 U/mg prot.); the difference was explained by the increased influence of particle mass transfer limitations. The immobilizates were stored for 18 months at 4 °C in a citrate–phosphate buffer (50 mM, pH 4.5) and the residual activity was measured. The immobilized enzyme activity was tested at regular intervals (for ten and later five consecutive repetitive hydrolysis reactions).

After each hydrolysis, the immobilizates were washed with a citrate–phosphate buffer and used for the next run. Fifty repeated reactions were performed, providing >99% hydrolysis in 90 min of reaction in each run (Figure 5). The biocatalyst retained about 90% of its initial activity in the final cycle. The results were in agreement with those published recently, showing the long-term stability of immobilized recombinant human Hex A on polylactic acid films [26]. The authors indicated an improvement in the stability and functionality of the enzyme in its immobilized form for over a year. Such results confirm that immobilization is a suitable technique for the long-term application of Hex. Moreover, it opens up opportunities for research and progressive biotechnological and biomedical applications of this enzyme.

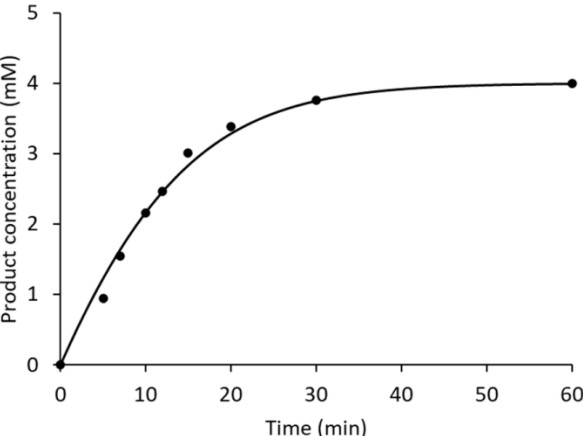

**Figure 4.** Kinetics of enzymatic hydrolysis of 4-NP-β-GalNAc catalyzed by hexosaminidase immobilized in LentiKats®. Points are experimental data; the solid line represents the data calculated using Equation (8).

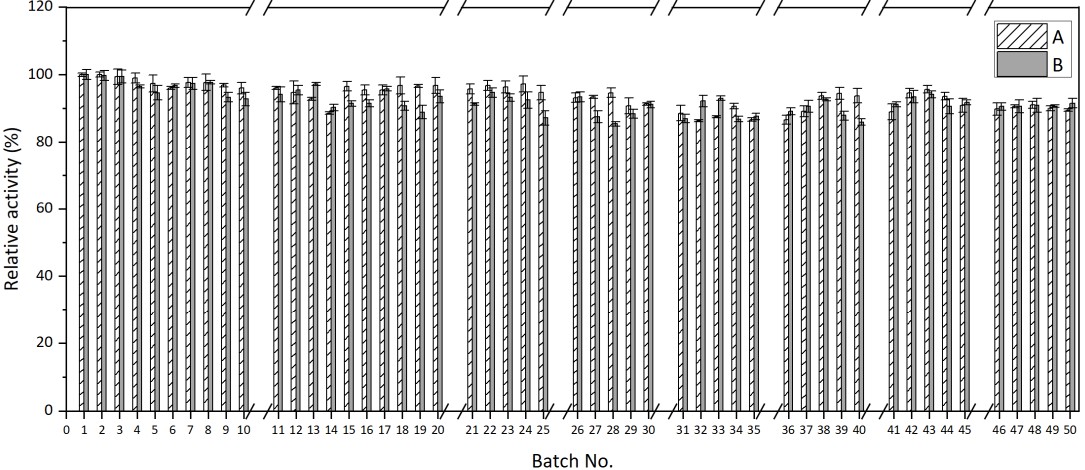

**Figure 5.** Repeated hydrolytic reactions with Hex immobilized in PVA-gel particles. Reaction conditions: 4 mL citrate–phosphate buffer (50 mM, pH = 4.5), 4 mM of the substrate 4-NP-β-GalNAc, 1 g of immobilizates, 35 °C, 300 rpm. 100% relative activity corresponds to 44 U/mg prot. (A) and 40 U/mg prot. (B).

## 3. Materials and Methods

### 3.1. Chemicals

4-nitrophenyl-2-acetamido-2-deoxy-β-D-galactopyranoside,4-nitrophenyl-2-acetamido-2-deoxy-α-D-galactopyranoside, 2-nitrophenyl-2-acetamido-2-deoxy-β-D-galactopyranoside, 4-nitrophenyl-2-acetamido-2-deoxy-β-D-glucopyranoside, 2-nitrophenyl-2-acetamido-2-

deoxy-β-D-glucopyranoside, and 4-methylumbelliferyl-2-acetamido-2-deoxy-β-D-galactopyranoside were purchased from Carbosynth Ltd. (Compton, UK). Methanol (HPLC Gradient grade) was purchased from Fisher Scientific UK Ltd. (Loughborough, UK). All other chemicals used in this study were of analytical grade and commercially available. All solutions were prepared using deionized water.

### 3.2. Microorganism and Cultivation Conditions

The β-*N*-acetylhexosaminidase producing strain *Penicillium oxalicum* CCF 1959 was obtained from the Culture Collection of Fungi (CFF), Department of Botany, Charles University in Prague (CZ). The cultivation medium contained (g/L): $KH_2PO_4$, 3.0; $NH_4H_2PO_4$, 5.0; $(NH_4)_2SO_4$, 2.0; yeast extract, 0.5; NaCl, 15.0; N-acetylglucosamine as the β-*N*-acetylhexosaminidase inducer, 5.0. The reaction pH was maintained at 6.0. After sterilization, a sterile solution of $MgSO_4$ (0.5 mL, 10% *w/v*) was added. The medium (100 mL) was inoculated with 1 mL of a suspension of spores ($6.0–7.0 \times 10^6$ /mL) in 0.1% (*v/v*) Tween 80 and cultivated in 500 mL flasks at 28 °C and 200 rpm (New Brunswick Scientific, Enfield, CT, USA) for 12–13 days.

### 3.3. Purification of β-N-Acetylhexosaminidase

The mycelium was removed from the culture medium by filtration, and the cell-free medium was used for enzyme purification. The enzyme was separated from the medium by cation exchange chromatography (Fractogel EMD $SO_3$– (Merck, Darmstadt, Germany)) using the ÄKTA Purifier protein chromatography system (GE Healthcare, Uppsala, Sweden) as described previously (Slámová et al., 2012). The purity of the fractions was checked on 10% SDS-PAGE. The enzyme was concentrated using an ultrafiltration membrane (Amicon® Ultra-4 10K, Merck, Kenilworth, NJ, SA) and stored in 1 M $(NH_4)_2SO_4$ at 4 °C with no considerable loss of activity for at least five months. Concentrations of protein stock solutions were 0.25–0.35 mg/mL depends on the individual enzyme batches. Protein was quantified using the Bradford assay calibrated for bovine serum albumin.

### 3.4. Immobilization in PVA Gel

The purified β-*N*-acetylhexosaminidase was immobilized by entrapment in PVA-based lenticular hydrogel particles (LentiKats®, Prague, Czech Republic) using the patented method [35]. The PVA gel solution comprised PVA (10% *w/v*), polyethylene glycol (6% *w/v*), and water. The solution was heated for 20 min at 90 °C in a water bath and then cooled to 40 °C under constant stirring. The purified enzyme solution (0.3 mL) was added to the gel (3.0 mL) and mixed for 5 min. Later, the particles were formed by dripping the solution onto a polystyrene plate using a syringe. The solution was dried in an airflow cabinet at 40 °C for 45 min and hardened in a stabilizing solution (0.1 M $Na_2SO_4$). The particles were washed with 50 mM citrate–phosphate buffer (pH 4.5) and stored at 4 °C or used directly for the hydrolytic reaction. The final average particle diameter was 2 mm.

### 3.5. Stability and Reusability of Immobilized β-N-Acetylhexosaminidase

The reusability and stability of the immobilized Hex were investigated within 18 months by repeated hydrolysis of 4-NP-β-GalNAc. The reaction mixture contained 25% (*w/v*) loading of the immobilized biocatalyst in 4 mM substrate solution in citrate–phosphate buffer (50 mM, pH 4.5). After each hydrolysis, the particles were washed twice in a 50 mM citrate–phosphate buffer (pH 4.5) and used in the next batch run or stored in the same buffer at 4 °C.

### 3.6. Enzyme Assay

β-*N*-Acetylhexosaminidase activity was measured spectrophotometrically using 4NP-β-GalNAc, 2NP-β-GalNAc, 4-NPGlcNAc, 2-NPGlcNAc, and 4-MUGalNAc as the substrates. One unit of enzyme activity was defined as the amount of enzyme releasing 1 μmol of nitrophenol (or 4-methylumbelliferone) per minute in 50 mM citrate–phosphate

buffer at pH 4.5 and 35 °C. The reaction mixtures were shaken at 450 rpm using the Eppendorf Thermomixer Comfort (Eppendorf, Cambridge, UK). Aliquots of 20 μL were withdrawn at various time intervals and added to 2.0 mL of 0.1 M $Na_2CO_3$. The released nitrophenol (or 4-methylumbelliferone) was determined spectrophotometrically (Eppendorf BioSpectrometer® basic, Eppendorf, Hamburg, Germany) at 420 nm (or 347 nm). All experiments were performed in triplicates.

The enzyme activities of β-*N*-acetylhexosaminidase were calculated from the linear part of a plot where hydrolysis was represented as a time function of the product concentration. The initial specific activity was calculated as the activity divided by the protein concentration in the purified enzyme solution used for hydrolysis. The initial specific activity of the immobilized enzyme was calculated by dividing the activity of the immobilized enzyme by the weight of the particles used for hydrolysis.

### 3.7. Preparation and Enzymatic Resolution of α/β-Substituted GalNAc Derivatives

An anomeric mixture was prepared from 3,4,6-tri-O-acetyl-2-azido-2-deoxy-α/β-D-galactopyranosyltrichloroacetimidates, which were converted to substituted 4-nitrophenyl-2-acetamido-2-deoxy-α/β-D-galactopyranosides [36]. In the next step, the azido group was reduced to the acetamido group by the Staudinger reaction [37] to yield 4-nitrophenyl-2-acetamido-2-deoxy-α/β-D-galactopyranosides.

For selective enzymatic hydrolysis, the synthesized mixture of 4-nitrophenyl-2-acetamido-2-deoxy-α/β-D-galactopyranozides (4-NP-α/β-GalNAc in the ratio of α:β anomers = 2:3), as well as anomeric mixture prepared from standard chemicals (anomeric ratio of α:β = 1:1) were used. The reaction mixture contained the substrate (4.4 mM solution in citrate–phosphate buffer pH 4.5, 18 mL) and the enzyme solution (4 U/mL, 2 mL). Hydrolysis was performed in 50 mL glass reactors at 35 °C for 60 min with stirring at 450 rpm. The reaction was terminated by heating to 100 °C for 5 min. After filtration, the reaction mixture was used for the isolation of α-substituted GalNAc derivatives.

### 3.8. Separation of α-Substituted GalNAc Derivatives

After enzymatic hydrolysis, 20 mL of the reaction mixture was extracted with 10 mL of the organic solvents (ethyl acetate and n-butanol). In the first step, nitrophenol was removed from the reaction mixture by extraction with ethyl acetate. After vigorous shaking, the phases were separated under gravity using a separatory funnel. The aqueous phase containing the α-substituted GalNAc derivative and N-acetyl-D-galactosamine was subsequently extracted with n-butanol. Thus, 98.5% of the α-substituted GalNAc derivative was extracted in the organic phase after three repeated extractions, and N-acetyl-D-galactosamine remained in the aqueous phase. The concentration of the compounds was determined by HPLC in both the top and bottom phases.

All experiments were performed in duplicate and the mean values were used for data treatment and presentation.

### 3.9. HPLC Analysis

The concentrations of both anomers of the GalNAc derivatives were measured by HPLC using the Agilent 1260 Infinity LC apparatus (Agilent Technologies, Waldbronn, Germany) with Poroshell 120 EC-C18 column (3 × 100 mm, 2.7 μm particle size) and Agilent EC C18 guard column (3 × 5 mm, 2.7 μm); the column temperature was maintained at 37 °C. The nitrophenyl derivatives of GalNAc anomers and nitrophenol were detected using the DAD VL+ detector (Agilent 1260, Santa Clara, CA, USA). The mobile phase constituted methanol/water (15:85, *v/v*), and the flow rate was maintained at 0.5 mL/min. The retention times were 9 min for the α-anomer of 4-NP-GalNAc, 6 min for the β-anomer of GalNAc, and 15 min for 4-nitrophenol. N-acetyl-D-galactosamine was analyzed by HPLC on the Watrex Polymer IEX H⁺ column (250 × 8 mm) (8 μm) (guarded by column with the same filling—50 × 8 mm, 8 μm). The flow rate was maintained at 0.7 mL/min, and

1.31 mM sulfuric acid served as the mobile phase at 45 °C; the detection was performed using the RI detector (retention time 11 min).

*3.10. Kinetic and Mass Transfer Characteristics of the Immobilized Hexosaminidase*

The kinetics of the reaction catalyzed by the soluble hexosaminidase was fitted by the Michaelis–Menten equation:

$$v = \frac{V_{max} c_S}{K_M + c_S} \tag{1}$$

where $v$ is the enzyme reaction rate, kinetic parameters are maximum reaction rate, $V_{max}$, and Michaelis constant, $K_M$. Practically, it was appropriate to rewrite $V_{max}$ as:

$$V_{max} = k_2 c_E \tag{2}$$

thus relating the $V_{max}$, enzyme concentration, $c_E$, and catalytic constant, $k_2$.

For the hexosaminidase immobilized in porous LentiKats® particles, the expected mass transfer limitations due to pore diffusion were expressed using the effectiveness factor, $\eta$:

$$\eta = \frac{v_o}{v_k} \tag{3}$$

where $v_o$ is the overall catalytic reaction rate observed in conditions of mass transfer limitation, and $v_k$ is the reaction rate calculated directly from the kinetic Equation (1). The effectiveness factor calculation requires the solution of differential equations involving the mass transfer and reaction phenomena, while the form of the equations depends on the particle geometry. However, the task can be simplified using solutions developed and published for Michaelis–Menten kinetics in terms of the modified Thiele modulus [38]:

$$\phi_m = \frac{b}{\sqrt{2}} \sqrt{\frac{V_{max}}{D_{eS} c_{Sp}} \left( \frac{1}{1+\beta} \right) \left[ 1 + \beta ln \left( \frac{1}{1+\beta} \right) \right]^{-1/2}} \tag{4}$$

where the parameter $\beta$ is a ratio of the Michaelis constant and substrate concentration at particle surface:

$$\beta = \frac{K_M}{c_{Sp}} \tag{5}$$

Equation (4) applies for infinity slab geometry, which is an acceptable approximation of LentiKats® particles similar to disks with a diameter of 20 times thickness. The other symbols in Equation (4) denote half a particle thickness, $b$, substrate effective diffusion coefficient, $D_{eS}$, and substrate concentration at the particle surface, $c_{Sp}$. Using the modified Thiele modulus, the effectiveness factor can be calculated with sufficient precision from the first-order kinetic approximation. For slab geometry and steady-state conditions (no particle accumulation), it is in the form [38]:

$$\eta = \frac{tanh 3\phi_m}{\phi_m} \tag{6}$$

After determining the effectiveness factor, the overall reaction rate can be calculated based on Equation (3):

$$v_o = \eta v_k \tag{7}$$

The value of $v_k$ can be directly calculated from the Michaelis–Menten equation introducing substrate concentration in the liquid phase. The immobilized hexosaminidase reaction was performed in conditions of a batch stirred reactor. The observed reaction rate calculation in such cases considers total immobilized particle volume, $V_p$, and volume of the liquid phase, $V_l$:

$$v_{obs} = -\frac{dc_S}{dt} = \frac{V_p}{V_l} \eta v_k \tag{8}$$

The application of Equation (8) requires a pseudo-steady-state approximation of the batch reactor conditions. This can only be accepted when the reaction dynamics is significantly slower than the mass transfer dynamics. Then, particle substrate accumulation has negligible influence on the kinetics of the observed substrate concentration change and it can be assumed that the substrate reaches the concentration profile rapidly in the particles close to the steady-state profile.

Equation (8) was numerically solved in Microsoft Excel 365 by a simple Euler method where the effectiveness factor was calculated at each time step. The value of the substrate (4-NP-β-GalNAc) effective diffusion coefficient in LentiKats® was taken from the published data [39] $3.6 \times 10^{-11}$ $m^2$ $s^{-1}$, which was the value determined for sucrose as it has the same molar weight. The average thickness of LentiKats® of $100 \times 10^{-6}$ m was measured using a micrometer.

## 4. Conclusions

In summary, an efficient method for the preparation of α-anomer of 4NP-α-GalNAc, a chromogenic substrate for the α-NAGA was successfully developed. The described process is based on a fungal β-*N*-acetylhexosaminidase, which selectively hydrolyzes the β-anomer of the glycoside and removes it from the aqueous solution or crystal suspension of the anomeric α/β-mixture. Different physicochemical properties allow the rapid and simple separation of all reaction products by solvent extraction. Moreover, glycone and aglycone components (products of β-anomer hydrolysis) can be easily recycled; thus, improving the attractiveness and cost-effectiveness of the overall process.

The immobilization of β-*N*-acetylhexosaminidase in PVA gel provided a stable biocatalyst and enabled its repeated use. Although the enzyme lost about 65% of its hydrolytic activity by immobilization, the total yield of the reaction was not affected. Storage of the immobilized enzyme at 4 °C for 18 months resulted in a 10% loss of the catalytic activity.

The described method has the potential for the preparation of other α-anomers of aromatic substituted glycosides as important chromogenic/fluorogenic substrates for the rapid diagnosis of various diseases or identification and quantification of enzymes.

## 5. Patents

The above process is patented under Slovak Patent Application no. 50042–2018 filed on 9 September 2018, and Slovak Utility Model no. 8652 with Application no. 50125–2018 filed on 28 November 2018.

**Supplementary Materials:** The following supporting information can be downloaded at: https://www.mdpi.com/article/10.3390/catal12050474/s1, Figure S1: The HPLC chromatograms of samples withdrawn from the enzymatic hydrolysis of 4-NP-α/β-GalNAc in crystal suspensions.

**Author Contributions:** Conceptualization, H.H. and V.Š.; methodology, H.H., V.Š., and E.O.; formal analysis, V.Š.; investigation, H.H., E.O., M.B., and J.V.; resources, J.V.; data curation, H.H., V.Š, E.O., and M.B.; writing—original draft preparation, H.H., V.Š., and E.O.; writing—review and editing, H.H. and V.Š.; supervision, H.H. and M.R.; funding acquisition, M.R. All authors have read and agreed to the published version of the manuscript.

**Funding:** This work was funded by the Agency for supporting research and development, according to the agreement Nr. APVV-20–0208.

**Data Availability Statement:** The data presented in this study are available on request from the corresponding author.

**Conflicts of Interest:** The authors declare no conflict of interest.

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
