# Peer review of "Chemo-Enzymatic Production of 4-Nitrophenyl-2-acetamido-2-deoxy-α-D-galactopyranoside Using Immobilized β-N-Acetylhexosaminidase"

_catalysts, doi:10.3390/catal12050474_

Round 1
Reviewer 1 Report
The paper "Chemo-enzymatic production of α-nitrophenyl derivatives of glycosides using immobilized β-N-acetylhexosaminidase" present a chemo-enzymatic preparation of an alpha-nitrophenyl glycoside substrate of alpha-NAGA, an enzyme with different applications in the biomedical field.
The paper is well organized an scientifically sounding and data are clearly presented, but it is not immediately clear the aim of the work and what's the value of preparing this nitrophenyl derivative, that is explained only at the end of the introduction. I would emphasize in the abstract that the final aim is to prepare a substrate for evaluating enzymatic activity of alpha-NAGA, that is a crucial enzyme for several biomedical application, and also I rearrange the introduction to make this more clear since the beginning.
minor issues:
- Figure 2: remove "-" after data symbols in the figure legend
- rows 246-247: the authors stated that the reduced activity upon immobilization 65% of the enzyme molecules are inactivated or inaccessible. Both hypothesis calls for an immobilization heterogeneity. Is it also possible that there is an homogeneous decrease of all enzyme molecules activity? The authors should comment on this, also based on the possibility to discriminate the two conditions in their experiments. The same consideration applied also to the end of the conclusion (rows 464-465).
- row 453: the term "biomedicine" should be referred to the enzyme alpha-NAGA rather than the prepared substrate, I would rephrase. Again, it is not clearly highlighted that the main result is the efficient production of chromogenic derivatives that can be used for testing an activity of an enzyme involved in a metabolic disorder and that can be used to prepare group zero RBCs.
Reviewer 2 Report
The authors produced α-nitrophenyl derivatives of glycosides by using free and immobilized β-N-acetylhexosaminidase to catalyze the β-anomer in the mixture of 4-NP-α/β-GalNAc. The system after reaction was extracted twice to remove the catalysates to get pure α-anomer. This work has provided a useful method to the preparation of α-nitrophenyl derivatives of glycosides from the anomer mixture, while the immobilization process of β-N-acetylhexosaminidase has also been investigated. I personally think that this manuscript could be accepted if the following questions are considered in the revised version.
- Line 49, the abbreviation b-NAGA should be given when β-N-acetylhexosaminidases appeared in the first time. After that, the abbreviation should be used instead of β-N-acetylhexosaminidases (line 52 etc.). Other abbreviations have the same problem.
- The ethyl acetate and n-butanol are used to extract the catalysates of 4-NP-β-GalNAc, how can you ensure the efficient or total separation of the two solvents? In section 2.4, the authors have only given the crystal purity of the 4-nitrophenyl-2-acetamido-2-deoxy-α-D-galactopyranoside, how is the yield?
- Section 2.4, how is the crystallization process performed since the final product is crystal?
- The authors have done much kinetic calculations on the enzymatic reaction according to 3.11, maybe it is more readable when the calculated parameters are all listed in the manuscript.
- This manuscript focused on the removal of β-anomer of the glycosides by enzymatic reaction from a mixture of α/β-anomers. Most of the content was about the enzyme and the enzymatic reaction, therefore I don’t think the word “chem-enzymatic” is suitable for this work. After all, the “chem” part was quite limited. So please reconsider the title.
Reviewer 3 Report
The manuscript of a research paper entitled “Chemo-enzymatic production of α-nitrophenyl derivatives of glycosides using immobilized β-N-acetylhexosaminidase” by Helena Hronska et al. submitted to Catalysts focuses on development of a biochemical method to synthesize 4NP-α-GalNAc (i.e. pNP-α-GalNAc), a selective chromogenic substrate for α-N-acetylgalactosaminidases. To achieve this a chemically synthesized precursor mixture was modified by immobilized fungal β-N-acetylhexosaminidase acting as the catalyst.
General remarks:
The subject of the manuscript is in a good accordance with the aims and scope of Catalysts as it describes an important method development for green chemistry. There are some novel aspects in the work as a new application of immobilized fungal β-N-acetylhexosaminidase was proposed to produce chromogenic substrate for biochemical assays of α-N-acetylgalactosaminidases. Still, the substrate is commercially available (CAS No. 23646-68-6) but with rather high price. The methods for producing chromogenic substrates with competitive prize and sustainable manner would be biotechnologically relevant.
The manuscript is generally adequately structured and composed, and mostly clearly presented but some additions or clarifications have to be made before the manuscript can be considered for publication. The manuscript is written in the correct academic style English language with minor amendments needed.
Specific comments:
Title. The title seems to be too broad for the presented study as it states “… α-nitrophenyl derivatives of glycosides” were produced. In the paper only one α-nitrophenyl derivative was obtained, i.e. 4NP-α-GalNAc, when α/β-substituted GalNAc derivatives were treated with β-N-acetylhexosaminidase. The same applies for the lines 96-98. It should be clarified how many α-derivatives and β-derivatives were in the mixture.
Lines 3, 10 and elsewhere throughout the manuscript. The “N” in N-linked compounds or the respective enzymes should be in italics, e.g. β-N-acetylhexosaminidase.
Line 15, 18, 19 and elsewhere. The correct abbreviation for β-N-acetylhexosaminidase would be β-NAHA, not β-NAGA.
Abstract. Line 10. Why the α-nitrophenyl derivatives of glycosides are efficient substrates? It would be more accurate to call them easily detectable or something similar due to the chromogenic reaction in alkaline conditions. Lines 14-15. It should be stated how many fungal 14 β-N-acetylhexosaminidases were tested (at least from the current wording suggests multiple enzymes) and which one was used for the reaction. Lines 20-21. The storage temperature should be added.
Introduction. The overall background and an overview are given on glucoside synthesis, fungal β-N-acetylhexosaminidases and the relevance of α-N-acetylgalactosaminidases. The Introduction is rather brief without some relevant background information regarding the β-N-acetylhexosaminidases and their properties of different origin and immobilization techniques of β-N-acetylhexosaminidases (or similar GH enzymes). Some specific examples of the enzyme characteristics and immobilization optimization relevant to the work could provide wider perspective of the study. The β-N-acetylhexosaminidases are very broadly distributed enzymes and hundreds of them have been characterized in addition to fungi and other eukaryotes also from bacteria. More information and the properties of some biotechnologically relevant β-N-acetylhexosaminidases should be provided.
On the lines 48-49 and 81-83 the information regarding the aim of the study is repeated twice. The aim of the work and the main rationale should be appearing in one place, preferably in the end of the Introduction.
Lines 48-52. The EC number (EC 3.2.1.52) and carbohydrate-active enzyme classification (GH families) of β-N-acetylhexosaminidases should be presented.
Lines 54-55. The important ability of some β-N-acetylhexosaminidases to transglycosylate is mentioned without further explanation of the context. More comprehensive information of the process involvement for biosynthesis should be given. It has to be noted that transglycosylation is also present in bacterial enzymes (relevant examples https://doi.org/10.1021/jf020965x; https://doi.org/10.1007/s00253-015-6550-0; https://doi.org/10.3390/ijms21020417).
Lines 87-89. The novelty statement could precede the summary of the work.
Results and Discussion. The Discussion is somewhat hindered and the part downsized compared to the extensive study which contain several different methodological approaches.
Lines 94-95. It should be clarified if the method was newly introduced or optimized for the 4NP-α-GalNAc synthesis. Relevant references should be added if appropriate.
Lines 115-116. The substrate for the obtained specific activity of the purified enzyme should be stated.
Lines 125-132. The discussion should be added to explain the importance of avoiding product-derived inhibition for enzyme immobilization.
Lines 127-129, 243-246. The determined constants/kinetics should be referred to the numbers of the equations shown in the Materials and Methods.
Fig. 2. Concentration of 6.48 mM seems to be missing some time points. Therefore, the fitting of the graph might not be very accurate.
Fig. 2, Fig. 3, Fig. 4, Fig. 5. How many technical parallel measurements were conducted to find average values presented on the figures? It should be indicated in the figure legends.
Line 147-148. How many enzymes were tested in the study? It seems from the manuscript that only one β-N-acetylhexosaminidase was prepared and applied.
Line 199. Is the obtained purity in standard range or exceptionally high for these kinds of processes? Relevant literature could be cited.
Lines 215-225. Which of the substrates have or should have the highest affinity and catalytic efficiency? Relevant references could be cited for comparison.
Materials and Methods. The section is generally comprehensive but some minor the experimental details should be clarified.
Lines 304-305. How much spores were inoculated to the medium?
Line 312-313. GE Healthcare (currently Cytiva) providing ÄKTA machinery is from Sweden.
Lines 315. What was the concentration of the protein stock solution? For some of the β-N-acetylhexosaminidases the diluted enzyme is less stable (e.g. https://doi.org/10.3390/ijms21020417).
Lines 326-327. What was the material of the used plate? What diameter droplets were prepared?
The short subchapters 3.7 and 3.8 could be combined.
Round 2
Reviewer 3 Report
Review to the revised manuscript newly entitled “Chemo-enzymatic production of 4-nitrophenyl-2-acetamido-2-deoxy-α-D-galactopyranoside using immobilized β-N-acetylhexosaminidase ” by Helena Hronská et al. submitted to Catalysts.
I sincerely thank the authors for their effort. The manuscript has been considerably improved during the revision.
The main issues and comments that were pointed out have been addressed by the authors and the necessary clarifications have been introduced to the manuscript, incl. the title and abstract. The updated abbreviation of the enzyme, i.e. Hex, is fully acceptable. The manuscript is in my opinion sufficiently updated to warrant publication.